# Transforaminal Epiduroscopic Laser Ablation for Removal of a Postlaminectomy Synovial Cyst: A Case Report

**DOI:** 10.3390/medicina56050209

**Published:** 2020-04-25

**Authors:** Hee Yong Kang, So Yeon Kim, Chung Hun Lee, Sung Wook Park

**Affiliations:** 1Department of Anesthesiology and Pain Medicine, Kyung Hee University Hospital, Kyungheedae Road 23, Dongdaemun-Gu, Seoul 02447, Korea; ujuabba@gmail.com (H.Y.K.); kimsso88@gmail.com (S.Y.K.); 2Department of Anesthesiology and Pain Medicine, Korea University Medical Center, Guro Hospital, Gurodong Road 148, Guro-Gu, Seoul 08308, Korea; bodlch@naver.com

**Keywords:** postlaminectomy, synovial cyst, transforaminal epiduroscopic laser ablation, laser, radiculopathy

## Abstract

*Background*: Synovial cysts rarely occur after a laminectomy and are difficult to detect if there are no symptoms; however, they can cause lower back pain or symptoms of radiculopathy. Various methods are used to treat synovial cysts. Here, we will introduce the first case with treatment using the transforaminal epiduroscopic laser annuloplasty (TELA) system. *Case report*: A 64-year-old female patient visited the pain clinic with lower back pain and pain radiating from the left lower extremity. An MRI T2 image showed a synovial cyst of facet joint origin at the L4–L5 level; the patient had undergone a laminectomy 10 years ago at the same spinal level. The patient rated the pain an 8 on the numerical rating scale (NRS), and pain was reduced after epidural steroid injection, but symptoms recurred a month later. The cyst ablation was performed using the TELA system with a 1414 nm neodymium-doped yttrium-aluminum-garnet (Nd:YAG) laser, and after the procedure, pain decreased to 4 points immediately and was reduced to 2 points on the NRS after 1 week. Six months after the procedure, the pain level was measured on NRS 2 and cyst was not recurred in the additional MRI. *Conclusion*: We introduced the TELA system as a noninvasive therapy for treating synovial cysts. Ablation of cystic necks using a 1414 nm Nd:YAG laser could be a method to prevent cyst recurrence, but long-term follow-up and large scale control studies will be needed to verify the effectiveness of this method.

## 1. Introduction

Spinal synovial cysts can be found incidentally, but typically present with lumbar pain or radiculopathy. There is evidence that they are associated with arthrosis due to instability of the facet joint and excessive load on the joint [1]. Although synovial cysts are considered benign, patients can present with radiculopathy and low back pain in association with spinal stenosis and/or, rarely, direct nerve root compression [2]. More rarely, synovial cysts occurring after spinal surgery have been reported [3,4].

Evidence for adequate treatment of symptomatic synovial cysts is lacking. Surgical removal is possible, but non-surgical methods are preferred, due to the invasiveness of the surgery and the burden of post anesthetic recovery. Non-surgical treatment methods, such as bed rest, physiotherapy, chiropractic treatment, acupuncture, oral analgesics, braces, and lumbar spine injections have been attempted [2]. Epidural steroid injections improved symptoms in approximately one-third of patients, but, for others, symptom improvement was temporary or insignificant [5].

Herein, we report a new method using the neodymium-doped yttrium-aluminum-garnet (Nd:YAG) laser of the transforaminal epiduroscopic laser annuloplasty (TELA) system to remove a synovial cyst that occurred after a laminectomy.

## 2. Case Report

The patient gave written, informed consent for publication. The study was conducted in accordance with the Declaration of Helsinki, and the protocol was approved by the Ethics Committee of Kyung Hee University Hospital (ethic code: 2020-04-062, approved date: 1 April 2020.). A patient visited the pain clinic with lower back pain and radiating pain from the left lower extremity that began a week prior. The magnetic resonance imaging (MRI) T2 image showed a synovial cyst of facet joint origin at the L4–L5 level (Figure 1A,B); the patient had a medical history of a laminectomy 10 years before at the same spinal level. The patient rated her pain an 8 on the numerical rating scale (NRS) where 0 is no pain and 10 is the worst possible pain. After epidural steroid injection, pain was 2 points, and the patient was treated with pregabalin 50 mg twice a day. However, 1 month after the procedure, the same symptoms recurred, and pain was rated at 8 points. The removal of the synovial cyst was planned using the TELA system.

The patient was in the prone position, awake, and under local anesthesia. After infiltration with 1% lidocaine, anteroposterior and lateral fluoroscopy was used to advance a spinal needle into the L5 suprapedicular notch at the Kambin’s triangle. This was performed through a 1 cm incision 14 cm left of the vertebral midline. A series of anteroposterior and lateral radiographs were performed throughout the procedure to ensure the correct trajectory. At the suprapedicular notch, 1% lidocaine was injected into the foraminal space. After sequential dilation, a beveled working cannula was placed below the superior articular process (SAP) (Figure 2A,B). A semirigid epiduroscope, NeedleView CH (Lutronic^®^, Ilsan, South Korea), was introduced through the working cannula. After debridement of the cyst with forceps, ablation was performed using an Nd:YAG laser set at 2.5 to 6 W. While monitoring the patient for nerve irritation, the laser ablation was performed using a side-firing Nd:YAG laser cable included in the TELA system (Lutronic^®^, Ilsan, South Korea) with an operating wavelength of 1414 nm. The patient was observed postoperatively for neurological deficits or other procedure-related problems and was discharged one day after the procedure with no complications.

The patient’s pain, measured with the NRS, decreased to 4 points immediately after the procedure and was 2 points after 1 week. Subsequently, at a 6-month follow-up, the patient was treated with pregabalin 50 mg twice a day, and pain control was well-maintained at 1–2 points. In addition, MRI was performed, indicating that cyst did not recur (Figure 3A,B).

## 3. Discussion

Synovial cysts are rare, and synovial cysts post-laminectomy are even more rare [3,4]. As a result, there is no clear course of treatment. If nonsurgical treatments do not work, surgical treatments such as a facetectomy or laminectomy may be performed. After surgical treatment, the pain is resolved considerably, but approximately 1.6% of patients experience side effects such as a dural tear, spinal nerve injury, epidural hematoma, seroma, cyst recurrence, or deep venous thrombosis [6]. Nevertheless, surgical resection has a recurrence rate of 11% after an average of 2 years, and surgical resection plus fusion has no recurrence rate for a follow up of 65 months [7]. Although surgery is effective for pain management, non-surgical treatment is often attempted, due to the cost of surgery and the risk of anesthesia and complications during surgery. Fluoroscopically guided lumbar synovial cyst rupture has been attempted as a more noninvasive treatment. Immediate pain was reduced in many patients but recurred later in 40–50% of patients at 1 year [8]. It is thought that recurrence occurs because this method disrupts the cyst but leaves the cyst neck intact.

One report used a side-firing holmium-doped yttrium-aluminum-garnet (HO:YAG) laser through endoscopy to remove a discal cyst [9]. The authors commented that laser-induced ablation could reduce recurrence and instability. Other advantages of side-firing laser ablation included its usefulness in small spaces associated with herniation, the method had less nerve root manipulation, and it caused less perineural fibrosis. Cyst recurrence was suspected to be the cause of the pain recurrence in our patient; therefore, based on the previous report, we chose to use laser ablation as a treatment to prevent the recurrence of the synovial cyst.

TELA is a recently developed method for patients with disc herniation with radiculopathy. TELA decompresses herniated discs using a laser or forceps. This procedure is usually performed using a side-firing 1414 nm Nd:YAG laser to decompress the herniated disc [10,11]. An Nd:YAG laser was shown to be easy to control, effective, and safe to use under the guidance of a spinal epiduroscope in a human cadaveric model [12]. Although Nd:YAG lasers were introduced earlier than HO:YAG lasers, HO:YAG lasers are most commonly used for ablation therapy, because the energy does not reach deep structures [10]. However, the 1414 nm Nd:YAG laser can be tuned to the wavelength of the HO:YAG laser, providing shallow tissue penetration compared to conventional lasers [13].

Safety and efficacy should be considered when using laser ablation. Ablation can be carried out with high energy lasers, but it is important to use an appropriate level of energy, as it can damage surrounding structures. One study reported that access to the cervical articular facet joints with an Nd:YAG laser had the potential to facilitate ankylosis procedures without affecting the spinal cord when 2000 J of energy was used [14]. In the porcine cadaveric study, 6 W delivery of a 1414 nm Nd:YAG laser did not raise the temperature to harmful levels [12]. In addition, histological findings showed no evidence of thermal damage to peripheral structures, including the spinal cord, spinal end plates, and the vertebral body. Due to the low energy and low exothermic nature of the Nd:YAG laser, the temperature around the resected lesion did not increase significantly, making the 1414 nm Nd:YAG laser easier to use in spinal procedures.

There is a limitation in this report, which is the fact that the follow-up period is short. In other paper, after removal of cyst, the degree of pain was followed up to 6 months [15] or other papers report recurrence of cyst within 6 months [16,17], but the recurrence of cyst was reported as an average of 24 months [7]. In addition, symptomatic fibrotic change in the area where cyst is removed will occur after 6 months. Our study suggested a way to remove synovial cyst, and to develop a way to prevent the recurrence of cyst, a comparative study of at least 2 years should be conducted to verify the effectiveness of our method.

To the best of our knowledge, there have been no reports on the use of a 1414 nm Nd:YAG laser for the removal of a synovial cyst. Forceps were used to debridement of the cyst and the laser was used to ablate the cyst neck to prevent recurrence. It is not clear how much laser ablation contributes to the complete removal of cysts to prevent recurrence. However, after the laser ablation, the symptoms improved for 6 months, while, after the epidural block, the symptoms recurred 1 month later. MRI is a definite way to check for recurrence of a synovial cyst, and MRI taken 6 months after the procedure showed no recurrence of cyst.

## 4. Conclusions

We introduced the TELA system as a noninvasive therapy for treating synovial cysts. Ablation of cysts using a 1414 nm Nd:YAG laser could be suggested as one of the ways to prevent recurrence of the cyst, but long-term follow-up and large scale control studies will be needed to confirm as an effective method.

## Figures and Tables

**Figure 1 medicina-56-00209-f001:**
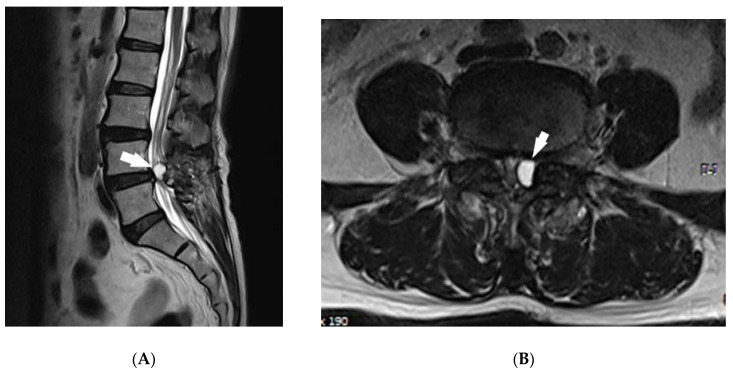
MRI with sagittal (**A**) and axial (**B**) T2-weighted images of the L4–L5 level, demonstrating a synovial cyst (white arrow) before procedure.

**Figure 2 medicina-56-00209-f002:**
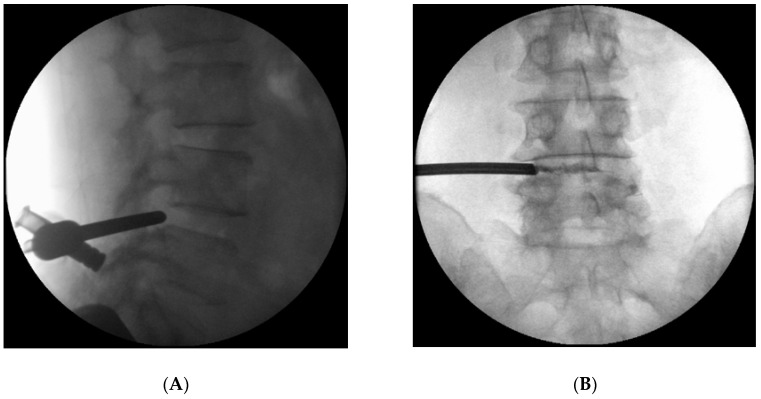
Lateral (**A**) and anteroposterior (**B**) views of the working cannula located in the superior articular process.

**Figure 3 medicina-56-00209-f003:**
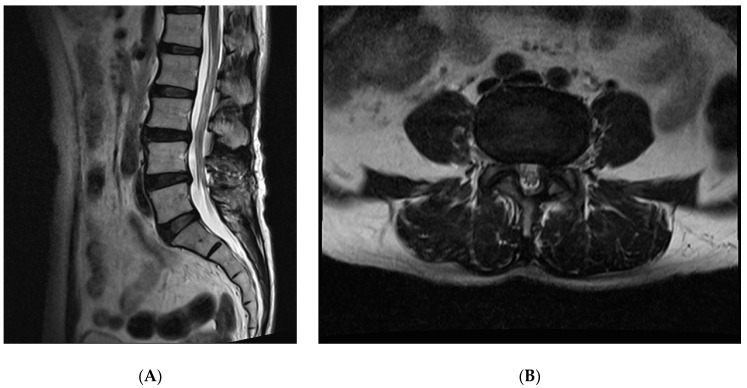
T2-weighted images of MRI with sagittal (**A**) and axial (**B**) taken 6 months after procedure.

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
