# Peer review of "Transforaminal Epiduroscopic Laser Ablation for Removal of a Postlaminectomy Synovial Cyst: A Case Report"

_medicina, 2020, doi:10.3390/medicina56050209_

Round 1

Reviewer 1 Report

The authors report on an novel method of non-invasive surgery for synovial cysts. The manuscript is well written and clearly structured. This case description is important for expanding the experience with this rare entity. The low pregabalin dose (100mg/d, 6 months postoperatively) indicates sufficient pain relief. The symptomatic fibrotic changes may also occur after 6 months. Perhaps a supplement in the manuscript can underline this.

Author Response

We are grateful to the reviewers for their insightful comments on my paper. Here is a response to the reviewers’ comments and concerns.

Reviewer 1

The authors report on an novel method of non-invasive surgery for synovial cysts. The manuscript is well written and clearly structured. This case description is important for expanding the experience with this rare entity. The low pregabalin dose (100mg/d, 6 months postoperatively) indicates sufficient pain relief. The symptomatic fibrotic changes may also occur after 6 months. Perhaps a supplement in the manuscript can underline this.

-Response: As reviewer pointed out, it is possible that a symptomatic fibrotic change will occur after 6 months. The limitation of our study is that the follow-up period is short. We have added content to the discussion.

Discussion

There is a limitation in this report, which is the fact that the follow-up period is short. In other paper, after removal of cyst, the degree of pain was followed up to 6 months [15] or other papers report recurrence of cyst within 6 months [16,17], but the recurrence of cyst was reported as an average of 24 months [7]. In addition, symptomatic fibrotic change in the area where cyst is removed will occur after 6 months. Our study suggested a way to remove synovial cyst, and to develop a way to prevent the recurrence of cyst, a comparative study of at least 2 years should be conducted to verify the effectiveness of our method.

Reviewer 2 Report

This is an well written interesting manuscript dealing with a difficult management.

The authors present no recurrence within 6 months after their transforaminal approach to remove a facet cyst with a forceps and coagulation of the cyst neck at the facet joint with a laser in an attempt to prevent recurrence.

The authors could add a paragraph on recurrence rates and other surgical approaches. eg percutaneous cyst rupture (Lutz 2017) of the cysts have a 40-50% recurrence rate at 1 year, but surgical resection has a recurrence rate of 11% after an average of 2 years, and surgical resection plus fusion has no recurrence rate for a follow up of 65 months (Wun 2019).

Thus, the weakness of the manuscript is the short follow up, which may prevent any firm conclusions to be drawn as the average time for cyst recurrence after surgical removal is 2 years.

In conclusion, either the authors may want to exclude recurrence after a longer follow up of 2 years, or turn the manuscript into a more technical report detailing their novel technique

Author Response

We are grateful to the reviewers for their insightful comments on my paper. Here is a response to the reviewers’ comments and concerns.

Reviewer2

This is an well written interesting manuscript dealing with a difficult management.

The authors present no recurrence within 6 months after their transforaminal approach to remove a facet cyst with a forceps and coagulation of the cyst neck at the facet joint with a laser in an attempt to prevent recurrence.

The authors could add a paragraph on recurrence rates and other surgical approaches. eg percutaneous cyst rupture (Lutz 2017) of the cysts have a 40-50% recurrence rate at 1 year, but surgical resection has a recurrence rate of 11% after an average of 2 years, and surgical resection plus fusion has no recurrence rate for a follow up of 65 months (Wun 2019).

-Response: As pointed out by the reviewer, the following changes were made to the discussion.

Nevertheless, surgical resection has a recurrence rate of 11% after an average of 2 years, and surgical resection plus fusion has no recurrence rate for a follow up of 65 months [7]. Although surgery is effective for pain management, non-surgical treatment is often attempted due to the cost of surgery and the risk of anesthesia and complications during surgery. Fluoroscopically-guided lumbar synovial cyst rupture has been attempted as a more noninvasive treatment. Immediate pain was reduced in many patients but recurred later in 40-50% of patients at 1 year [8]. It is thought that recurrence occurs because this method disrupts the cyst but leaves the cyst neck intact.

Thus, the weakness of the manuscript is the short follow up, which may prevent any firm conclusions to be drawn as the average time for cyst recurrence after surgical removal is 2 years. In conclusion, either the authors may want to exclude recurrence after a longer follow up of 2 years, or turn the manuscript into a more technical report detailing their novel technique

-Response: As the reviewer pointed out, there is a limitation in our report with a short patient follow-up period. However, there are some papers that have compared changes in pain up to 6 months after removal of cyst or reported recurrence of cyst within 6 months, so we performed an additional MRI after 6 months. However, as the reviewer pointed out, the follow-up period is short, so there is a lack of a method to prevent the recurrence of cyst. The following contents were revised in the discussion about these limitations.

Discussion

There is a limitation in this report, which is the fact that the follow-up period is short. In other paper, after removal of cyst, the degree of pain was followed up to 6 months [15] or other papers report recurrence of cyst within 6 months [16,17], but the recurrence of cyst was reported as an average of 24 months [7]. In addition, symptomatic fibrotic change in the area where cyst is removed will occur after 6 months. Our study suggested a way to remove synovial cyst, and to develop a way to prevent the recurrence of cyst, a comparative study of at least 2 years should be conducted to verify the effectiveness of our method.

Conclusion

We introduced the TELA system as a noninvasive therapy for treating synovial cysts. Ablation of cysts using a 1414 nm Nd:YAG laser could be suggested as one of the ways to prevent recurrence of the cyst, but long-term follow-up and large scale control studies will be needed to confirm as an effective method.

Round 2

Reviewer 2 Report

the revision part requires some language editing